# A Scoping Review of the Factor Associated with Older Adults’ Mobility Barriers

**DOI:** 10.3390/ijerph20054243

**Published:** 2023-02-27

**Authors:** Nur Hasna Che Had, Khadijah Alavi, Noremy Md Akhir, Irina Riyanti Muhammad Nur, Muhammad Shakir Zufayri Shuhaimi, Hui Foh Foong

**Affiliations:** 1Social Work Program, Faculty of Social Sciences & Humanities, Universiti Kebangsaan Malaysia, Bangi 43000, Selangor, Malaysia; 2Malaysian Research Institute on Ageing (MyAgeingTM), Universiti Putra Malaysia, Serdang 43400, Selangor, Malaysia

**Keywords:** mobility, barriers, challenges, elderly well-being, sustainable development goals (SDGs)

## Abstract

The phenomenon of ageing may contribute to the rise of the dependent population. Due to the obstacles and difficulties they confront, the elderly’s mobility decreases significantly. The aim of this article is to identify factors associated with mobility barriers in older adults. The method employed is an examination of articles published between 2011 and 2022 to identify common themes in previous studies. Four search engines were being used, and 32 articles have been included. This study demonstrated that health is a major factor associated with decreased mobility. This review identified four types of barriers which are health, built environment, socio-economic background and social relation change. This review could help policy makers and gerontologist in identifying solutions to resolve the mobility issues in older people.

## 1. Introduction

The steady increase in older individuals has caused much global concern. Malaysia is projected to encounter the ageing phenomenon by 2030 with an elderly population of 15.3% [1]. Mobility significantly influences older adults’ well-being and independence [2] as part of the fundamental needs of ageing [3]. Regardless, the elderly experience a decline in physical and psychological abilities during the ageing process [4]. As such, ageing is associated with low mobility levels, which worsens with disability [5]. Older adults reflect fewer trip frequencies [6,7], reduced travel time [8] and minimal diverse and outdoor activities [9].

Mobility denotes the ability to move across places and outside the home in daily life [10], and select (i) where and when to travel and (ii) which activities to engage in [11]. Older individuals require mobility to fulfil their basic needs. Carp’s conceptual model indicates food, clothes, medical services and money as living needs, while socializing, leisure, religious activities and recreation as older adults’ additional needs [12]. In this model, mobility is fundamental in determining life-sustaining and additional needs [3]. Both requirements deliver a significant contribution to well-being in later life. Stanley and Lucas indicated mobility as a tool to attain Maslow’s hierarchy of needs [13]. In this manner, mobility determines the maintenance needs in life [14] for optimal well-being. The desire for independence, control, status retention, integration, ‘normalcy’, and travel for its own sake, underpin older people’s intention to travel. The relevance of mobility goes beyond accessibility. All these factors influence people’s impression of their quality of life [15].

Studies show that, a decrease in outdoor participation among the elderly contributes to loneliness [16]. Loneliness has been identified as a major mental health problem among the elderly, and it can occur when social networks deteriorate [17] due to loss and life changes [18]. Almost half of the elderly in Peninsular Malaysia, specifically urban-area residents, are at risk of isolation [19]. Such loneliness could be exacerbated by low social support, physical disability, and chronic ailments [18]. Older adults must be assisted in coping with loneliness for a high quality of life. Overall, mobility is an effective intervention against isolation and loneliness [15,20], which lowers depression-related risks [21].

The barriers that the elderly face in implementing mobility contribute to their decline in outdoor mobility. Following past works, considerable disparities were identified in outdoor mobility due to the following factors: retirement [22], the ability to drive [23,24,25], having a companion [24], household income [23], and health conditions [23,25]. In line with specific studies, mobility differences were evident between urban and rural areas [26,27], with an emphasis on infrastructure and facilities [28], specifically public transportation [29,30]. Haustein and Siren’s literature review from 2000 to 2010, which encompassed the European region, identified the factors differentiating older adults’ mobility: health, gender, socio-economic background, car availability, environment, and social networks [31]. Notwithstanding, the distinguishing elements of mobility patterns and behaviours could not be classified under factors associated with mobility barriers. This mobility barrier implies the factors preventing older individuals from achieving the desired mobility and frequency [32].

## 2. Materials and Methods

### 2.1. Research Question

This article focuses on older adults’ mobility-oriented barriers and challenges. It is deemed crucial to identify such barriers to resolve these mobility issues. The research question of this review is presented as follows: “What are the factors associated with mobility barriers among older adults?”.

### 2.2. Search Strategy and Inclusion Criteria

This scoping review was conducted by screening four electronic databases between 2011 and 2021 through Scopus, Web of Science (WoS), Taylor & France Journal and SAGE, to identify a substantial number of published works. The study was confined to the 2011–2022 range for a broad spectrum of recent research on significant topics. A keyword combination (elder OR older adult AND travel OR mobility AND barrier OR difficulty OR challenges) was sought in titles, abstracts and keywords, to ensure that the article search results fulfil the study purpose based on the screening process criteria. The search was refined in relevant English-language journal categories, such as gerontology, social science, psychology, art and humanity, social work, or transportation. Notably, the records were screened with titles and abstracts to identify relevant articles following the keywords.

### 2.3. Screening

Full-text journals were retrieved and screened to determine the inclusion eligibility. The Preferred Reporting Items for Systematic Reviews (PRISMA) protocol was used to screen these articles [33,34]. The use of PRISMA has several advantages, including the ability to plan in advance, regarding the purpose and form (such as criteria), avoiding duplication, allowing articles to be systematically filtered through predetermined criteria, and enabling other parties or researchers to review, copy, or compare the results protocol [35].

The search strategy generated 6084 searches. A total of 5960 irrelevant articles were removed, while 124 were eligible for full-text reading based on the title or abstract. The eligibility criteria for mobility barriers are presented as follows: (i) research participants must include older adults (>60 years old); (ii) participants’ mobility-oriented barriers must be discussed; (iii) research rather than a review study must be conducted. Such criteria were established to ensure that the elicited results significantly demonstrated the key determinants of older adults’ mobility barriers. Summarily, 26 articles were selected for data extraction upon being skimmed. Figure 1 depicts the PRISMA process.

### 2.4. Data Analysis

Thematic analysis (TA), which enables the systematic evaluation of big data [36,37], was used in this study by generating specific theme patterns, following the researchers’ aims. Essentially, TA could clearly depict the similarities and differences between all the study datasets [38]. This analysis also indicates the frequency with which a pattern occurs [38], thus increasing the research accuracy, complexity, and significance. The current study articles were systematically read, with all the statements on older adults’ mobility barriers duly coded. Subsequently, the results were analysed by categorising them into multiple sub-themes. Pertinent sub-themes were then merged into specific themes [36].

## 3. Results

Eighteen out of the 32 studies were performed in Europe, six in North America, six in Asia, and two in Australia. Various mobility barriers among older adults were identified based on the data extraction outcomes. Two article types were identified: (i) quantitative and (ii) qualitative research. Twenty studies utilised the quantitative method, while 12 employed the qualitative approach. Table 1 depicts the themes derived from TA-based studies. The emergent themes from this review include: (i) health and disability; (ii) build environment; (iii) socio-economic background; (iv) social-relation changes; (v) weather. Table 2 summarises the articles containing the study purpose and their primary outcome.

### 3.1. Health, Disabilities and Fear of Falling

As documented in 19 articles [11,25,39,40,41,42,43,44,45,46,47,48,49,50,51,52,53,54,55], poor individual health instigates restricted mobility. One’s health condition induced specific disabilities, such as difficulty in walking [49], the use of a mobility device (wheelchair or walking stick) [48], and reliance on others [50]. As most elderly drivers stopped driving, owing to health concerns [40,45], older adults with disabilities had fewer transportation alternatives [47]. Mobility limitations owing to poor health also instigated the fear of falls or accidents among older adults [50], which prevented them from engaging in outdoor activities [48].

### 3.2. Build Environment

Housing location is a key determinant of the current state of facilities. Long distances from one’s home to the nearest public transportation [46,51,56] and environmental factors (hills, stairs, and parking availability) could discourage elderly people [16,51,52,56,57]. These barriers denote the built environment quality [49]. The availability of local amenities (food stores, newsagents, and post offices) influenced their motivation to venture out. Elderly drivers also complained about poor road structures and were confused about road signs, thus increasing their driving difficulties [58]. Land development might also adversely impact older individuals’ mobility. The urbanisation process could alter the original environment [49] and cause elderly individuals to feel less connected to the place.

Seven articles addressed the inconvenience of public transportation for older adults. Most older adults discovered that public transportation is unsuitable for them due to health issues and disabilities. For example, elderly people struggle to board and disembark from public transportation [50] and walk a long distance to the bus stop or train station [42,50,52], following limited access to public transportation [11,52]. Such distances also consume much travelling time [59]. Inconvenient schedules, unsuitable routes, and low frequency discouraged these individuals from using public transportation [50,59,60]. Older users also complained about the discomfort of public transportation shelters, drivers’ and other people’s behaviour [58], lack of assistance, and unsafe environments [61].

**Table 2 ijerph-20-04243-t002:** Article summary.

Author	Country/Setting	Objective	Design/Method	Sample Size	Characteristics	Analysis Method	Finding
Nordbakke and Schwanen 2015 [11]	Norway	To investigate the relationship between transportation and well-being by examining the extent to which older adults believed their needs for outdoor activity participation were unsatisfied.	Survey	4712	Aged 67 and above	SPSS	The level of unmet needs for out-of-home activities was shaped by transportation-related factors, such as having a driver’s license and subjective evaluations of public transportation supply. Actual participation in out-of-home activities, self-perceived health, and walking problems, outlook on life, residential location, and indicators of social support and networks, also explained differences in the extent of unmet activity needs.
Rantakokko et al. 2017 [16]	Finland	To perceive the relationship between perceived environmental barriers to outdoor mobility and loneliness among a community-dwelling older people.	Survey	848	Aged 75–90 years	Wald test by applying the delta method	Long distances to services and nearby hills, directly or indirectly, increased loneliness through restricted autonomy in outdoor participation.
Siren and Haustein 2016 [22]	Denmark	To observe how retirement affected older adult travel.	Telephone interviews	864	Born in 1946 to 1947	Pearson’s χ^2^ test, Kruskal–Wallis H-test and variance analysis	Owing to retirement, a clear tendency to reduce car use and mileage were highlighted.
Yang et al. 2018 [23]	United States	To examine active travel and public transportation used among older adults, and the built environment characteristics associated with them.	Survey	180,475	Aged 45 and above	Linear regression models and logistic regression models	Older adults over the age of 75 made fewer total trips, had lesser variety in travel purposes, and travelled shorter distances. Female elderly with medical conditions, who did not drive and had a lower household income tended to make fewer total trips, reflected a lower diversity of trips and travelled a shorter distance.
Shirgaokar et al. 2020 [25]	Canada	To investigate older adults’ unmet travel needs, and the relationship between personal abilities, living situation and socio-demographic factors with the trips not taken. To compare the likelihood of trips not taken following the lack of a ride in urban versus rural areas.	Survey	1390	Aged 65 and above	Ordinal logit models	Compared to older adults in urban areas, older adults in rural areas tended not to travel as they lived alone or in low-density housing.
Berg 2016 [39]	Sweden	To explore how mobility strategies evolved in the first years of retirement.	Interviews	27	Aged 66 to 73 years and retired	Content analysis	During the first year of retirement, significant changes involving illness or a decline in physical and social networks, and changing residence impacted, mobility strategies.
Choi and DiNitto 2016 [40]	United States	To investigate alternative modes of transportation used by non-driving older adults and their impact on well-being.	Survey	12,093	Aged 65 and above	Stata13/MP’s svy	Non-drivers relied on their informal support system and/or paid assistance to travel. Health deterioration was the most common cause of driving cessation.
Corran et al. 2018 [41]	London	To investigate the indicator of immobility in later life.	Travel diary data	123,562	Aged 18 and above	Logistic regression model	Retirement and disability were significant contributors to mobility decline.
Hjorthol 2013 [42]	Norway	To investigate the distribution of transportation resources among various groups of older people, unmet transportation needs, and their relationship to their well-being.	Survey	4723	Aged 67 and above	SPSS	Health, age, and transportation resources (driver’s license and access to a car) significantly impacted the unmet need to visit others, whereas gender and place of living demonstrated no effect.
Luiu and Tight 2021 [43]	England	To investigate the factors contributing to travel difficulties among people over the age of 60.	Survey	4025	Aged 60 and above	SPSS (descriptive statistics and binomial logistic regressions)	Poor health and well-being, lack of transportation resources, and gender were the main predictors of experiencing travel difficulties later in life. Travel proved more difficult for older people who lived alone or are widowed.
Mariotti et al. 2021 [44]	Italy	To explore the motivations of older adults in Milan and Genoa to not take trips and activities, owing to the perceived inadequacy of public transportation.	Survey	411	Aged 65 and above	Multivariate logistic regression models	Age, gender, and other control factors were the most significant variables associated with health status, neighbourhood, and LPT satisfaction. Furthermore, perceived LPT service quality and neighbourhood satisfaction influenced the likelihood of abandoning trips and activities: higher satisfaction induced lower likelihood of abandonment.
Murray and Musselwhite 2019 [45]	United Kingdom	To investigate the experiences of people who have stopped driving with informal support, following their decision based on individual circumstances.	Semi-structured in-depth interviews	7	Aged over 60 years and given up driving within the previous 6 years	Thematic analysis	Physical health issues were the primary reason for quitting driving, which also rendered it impossible to walk or use public transportation. When receiving lifts from family, friends, and neighbours: cars, the element of personal assistance and the accommodation of retired drivers’ physical mobility needs were recognised as important factors.
Noh and Joh 2012 [46]	South Korea	To examine elderly travel patterns in Seoul, South Korea.	Survey	481	Aged 65 and above	Sequence alignment method	Older age, living alone, a high level of physical disability, a low level of education, long distances from home to the nearest public transportation, having paid work, and the inability to drive discouraged the elderly from travelling.
Ryan et al. 2019 [47]	Sweden	To determine which resources and characteristics were associated with fewer opportunities among those aged between 65 and 79 years compared to their peers.	Survey	1149	Aged 65–79	Statistical analyses	Travelling proved complicated due to health issues. Income significantly impacted how people perceived their health.
Smith et al. 2016 [48]	Detroit	To investigate the impact of individual and community risk factors on mobility trajectories in a vulnerable community-dwelling elderly population.	Survey data	1188	Aged 55 and above	Latent-class growth analysis	Older age, severe mobility impairment, and the fear of falling were risk factors for membership in homebound and infrequent-mobility groups. Being homebound was associated with outdoors barriers.
Stjerborg et al. 2015 [49]	Sweden	To identify the daily changing mobility of an elderly couple living in a Swedish suburb.	Semi-structured interviews and time-geographical diaries	2	Older couple (married)	Narrative	Older adults were highly dependent on car use. The deterioration of health impacted their mobility ability, and surrounding barriers and authority constraints.
Faber and Van-Lierop 2020 [50]	Netherland	To investigate older adults’ mobility needs and desires in the Dutch province of Utrecht, and assess how they envisioned the future use of four different AV scenarios.	Focus group discussion	24	Older adults	Content analysis	Elderly perceived barriers to using active modes, such as walking, cycling, and public transportation, due to mobility limitations or fear of an accident.
Scott et al. 2023 [51]	Australia	To analyse the frequency of several personal and environmental obstacles.	Telephone survey	432	Aged 65 years old or older	-	Physical health was the most frequently reported impediment, followed by sensory issues, financial constraints, and caregiving obligations.
Luoma-Halkola and Haikio 2022 [52]	Finland	To explore older individuals’ perspectives of how they manage outdoor mobility and independent living when faced with mobility limitations.	Focus group interview	28	Older people	Thematic analysis	The elderly encountered mobility limitations owing to personal health issues and a wide range of contextual factors (inclement weather, lengthy travelling distances, hills, loss of local amenities, construction projects, spousal disease, and institutional aged care and health-care settings).
Gong et al. 2022 [53]	China	To identify the barriers to community care access in senior-only urban households.	Phenomenology approach using in-depth interview	18	Elderly aged 75 and above	Content analysis	Older persons frequently suffered from multiple chronic conditions that hindered their physical access to care resources.
Dickins et al. 2022 [54]	Australia	To determine the barriers to and facilitators of service access for this population.	Semi structured interview	37	Elderly women living alone	Thematic analysis	Health was the leading cause of women’s loss of driving privileges, with serious health occurrences precipitating licence revocation. Many participants mentioned that friends and family drove them to engagements; this dependence was usually prefaced by apprehensions and a desire not to disturb them.
Kuo et al. 2022 [55]	Taiwan	To analyse the risk factors associated with the longitudinal course of mobility problems and falls.	Data from Taiwan longitudinal study on aging (2003–2015)	5267	Middle-aged and older adults	Linear mixed-effects regression models and cumulative logit model analysis	The elderly reported having difficulty standing, walking, kneeling, and jogging. The likelihood of repeated falls, the amount of mobility impairment, cognitive status, living alone, and the number of comorbid conditions rose considerably with age.
Nordbakke 2013 [56]	Norway	To examine older women’s daily travel needs, behaviours, and activity participations in an urban setting, and investigate the complex relations between barriers, strategies and alternatives for mobility in old age.	Focus group interviews	31	Women aged 67–89	Thematic	Individual resources, contextual conditions, and strategies were interconnected, thus resulting in the opportunity for mobility.
Ozbilen et al. 2022 [57]	Ohio, US	To explore elderly travel patterns with an emphasis on the elements leading to sustainable mobility patterns.	Survey data	1221	Aged 60 years or older	Multinomial logistic regression model analysis	In mid-sized, and auto-dependent metropolitan areas, enhancements to the built environment supported sustainable travel among the elderly.
Misfud et al. 2019 [58]	Malta	To explore the psychological factors influencing older people’s mobility in Malta.	Survey	500	Aged 60 and above	Structural equation modelling	Older people were uncomfortable with public transportation. Their health issues also limited their ability to travel.
Mattson 2011 [59]	Dakota	To investigate ageing and mobility problems in rural and small urban areas.	Survey	1009	50–97 years old (AARP member)	Logit model	Public transportation was an option for the elderly who could not or did not intend to drive, but several barriers or problems discouraged their use.
Luiu et al. 2018 [60]	Birmingham	To examine the factors influencing elderly travel needs.	Survey	288	Aged 60 and above, live in urban area	IBM SPSS Statistics 24	Car ownership and individuals’ health and wellbeing were the two primary factors influencing travel need fulfilment.
Sundling et al. 2016 [61]	Sweden	To determine how negative or positive critical incidents in the public transportation environment affected behaviour, and examine how travel behaviour had changed.	In-depth interview	30	Older adult aged 65–91 and experiencing public transport	Qualitative method	Some cases negatively impacted on travel behaviour. Most critical incidents occurred in the physical environment of vehicles and stations/stops, and pricing/ticketing.
He et al. 2018 [62]	Hong Kong	To understand the impact of the economy on elderly mobility.	Survey	47,794	Aged 18 and above	Descriptive statistics	Some seniors with certain socioeconomic and geographic characteristics encountered potential spatial barriers to fulfil their mobility needs at certain times of the day.
Siren et al.2015 [63]	Denmark	To explore the relationship between mobility and well-being by emphasising various types of everyday out-of-home activities.	Semi-structured interviews	11	Aged 80–95 (experienced mobility-related limitations)	Qualitative method	With increasing mobility impediments, older adults’ prioritised and selected their activities for only necessary and nearby activities.
Ahmad et al. 2019 [64]	Pakistan	To understand elderly individuals’ current mobility characteristics, perceived needs, and limiting factors.	Survey	450	Aged 60 years or older	Descriptive and comparative analyses	Vehicle ownership and socio-demographic factors significantly impacted trip-making. Older people were concerned about public transportation and self-driving safety, and the behaviour of transportation crews.
Kim et al. 2014 [65]	South Korea	To investigate transportation deficiencies for older adults in Seoul.	Survey	812	Aged 65 and lived in Seoul	Ordered logit model	Low-income-earning participants who were 75 or older, with a physical disability, who had given up driving and lived with children in areas with difficult pedestrian conditions, might have limited access to transportation.

### 3.3. Socio-Economic Background

Wealth, education, and retirement were among the socio-economic aspects highlighted. Older adults with better socio-economic [23,51,62] and educational backgrounds [46] demonstrated high mobility. Retired seniors depicted low mobility [63], as they had fewer reasons to leave the house [41]. This pattern inevitably impacted their spouse [39]. Meanwhile, elderly individuals who lived in affluent neighbourhoods reflected a very high frequency of leisure trips [46]. Thus, income played a pivotal role in perceived health conditions, where higher-income earners observed greater transportation alternatives [47].

Affluence similarly influenced the possession of a vehicle and driver’s license. Five articles addressed the impact of owning a car and a driver’s license on older adults’ mobility. Being a driver and owning a car increased the availability of convenient, comfortable, and optimal [60] elderly transportation resources [42]. In this sense, vehicle ownership significantly affected elderly trip-planning [42,64]. Seniors who could drive themselves reflected a much higher frequency of leisure trips [46], while those who could not depended on family members or friends [56].

### 3.4. Social Relation Change

The death of a spouse, relative, or acquaintance might induce loss of companionship and dependability. Eleven articles explored how the loss or absence of a loved one could instigate mobility isolation. The presence of another elderly companion at home demonstrated a significant and positive impact on individual mobility [43,54]. Parallel to past studies, losing a spouse or companion [39,49], or living alone, causes mobility impairment [46,55] and transport deficiency [39]. The absence of an acquaintance may also be caused by the relocation of homes [39]. Being in one place for an extended period strengthens elderly people’s social network and support (companion, spouse, family, and friends) and reduces mobility deficiency risks [65]. Older people, particularly women, who generally refrain from travelling alone [66], could seek assistance from their social network or the larger community to travel and participate in outdoor activities [11].

### 3.5. Weather

Only two articles highlighted older adults’ difficulties in terms of weather conditions [16,52]. Following Rantakokko, the elderly would limit their outdoor mobility in extreme weather conditions, such as heavy snow [16].

## 4. Discussion

Luiu et al. investigated the factors associated with unmet travel needs, which were divided into three categories: health, transportation, and non-transportation [32]. Transportation issues were extensively discussed. Mollenkopf et al., who examined the factors influencing rural and urban elderly mobility behaviours, created a mobility model that identifies personnel resources, socio-economic resources, and structural or regional resources as contributors to outdoor mobility [67]. Regardless, this study classified personnel and socio-economic resources under the same group following their depiction of both individual resources and limitations. From a social ecology perspective, Yeom et al. addressed the intrapersonal, interpersonal, environmental, and organisational risk factors associated with mobility limitations [68]. Based on Yeom et al., interpersonal elements include gender and education level. These factors did not hamper elderly mobility unless they influenced other factors, such as socio-economic status. Thus, the TA outcomes could be divided into two-factor categories: personal (health, social change, and socio-economic background) and environmental (built environment and weather) factors.

### 4.1. Personal Factors

Low elderly mobility primarily results from health issues. Following Alavi, ageing caused physical and emotional changes that are inextricably linked to diseases [69]. An older adult’s quality of life is frequently related to health problems, movement difficulties, physical abilities, and sudden emotional shifts. Ageing potentially causes various physical and mental changes and the ability to act and move. These changes could reduce older individuals’ well-being. The ageing process constitutes optimal, normal, and pathological ageing [4]. The elderly would inevitably face minimal loss of physical function and health, even in optimal ageing, which worsens with pathological ageing [4]. Ageing, injury, and illness eventually lead to the reduction of or inability to perform specific activities [70].

Low mobility is also caused by poor mental health. Personality changes, mental functioning, and individual sanity indicate psychological ageing [4]. The decline in mental abilities is more complicated than their physical counterparts. Dementia, Alzheimer’s disease, and depression are frequently associated with reduced psychological ability. For example, dementia patients often experience depression, anxiety, and confusion, which hampers the performance of daily activities [71].

Older adults grow dependent on their social network owing to poor health and low capacity. Spouses, family members, and friends constitute an elderly person’s social network and support. Such networks encourage older adults with limited mobility to engage in activities [72]. Based on the statistics disclosed by the United Nations in 2019, the number of elderly people living alone or with only one partner proved insignificant. In 2010, 27% of the senior population in Asia resided alone or with a spouse [73]. Hamid et al. added that 10% of the 2322 elderly respondents in Peninsular Malaysia lived on their own [74]. These individuals required assistance from the surrounding community or appropriate facilities and services.

In line with Kim et al., education level correlated to income and vehicle ownership, which increased non-home activities [65]. Numerous studies correlated wealth to health status. Highly educated older adults demonstrated better knowledge and cognition, which led to optimal health behaviour and usage of medical services [75]. As such, higher-income earners tended to enjoy better health, independence, a longer lifespan [75], and various transportation options that mitigate outdoor mobility barriers.

### 4.2. Environment Factors

The environment, which significantly affects elderly mobility, could be divided into natural barriers and facilities or structures. Bad weather and hills could be a natural environmental challenge for the elderly to perform outdoor mobility. Facilities and structures involving stairs, poor road lighting, long distance to transportation services [16], and public transportation (buses and trains) are also deemed inconducive for older adults.

Older people struggle to board buses and trains. Several studies elaborated on transportation availability, accessibility and connectivity [7,20,42,60,76,77,78,79]. As most public transportation routes focused on getting people to work [76], the route and boarding time proved unsuitable for older people’s destinations. Other discomforts, such as the crowd, lack of hygiene [7,15,20], and boarding issues [7,60,61], also signified primary issues.

Elderly drivers worried about road conditions, insufficient rest areas, curb height, and inadequate lighting [16]. These individuals become confused following the urbanisation process, which inevitably altered the physical environment. Traffic congestion and noise pollution could also make them feel uncomfortable, unsafe, and uneasy. The urbanisation process has caused older people to lose place attachments, such as the emotional bond with the environment that fulfils essential human needs (comfort, happiness, safety, and security) [80] or the emotional man–landscape connection [81].

### 4.3. Recommended Strategies

The aforementioned issues must be resolved, as older adults should not be home-bound, regardless of the reasons underlying their low mobility. Changes in their social environment, including family functions [82], social support, social networks, mortality, financial concerns, and chronic diseases, would induce depression [83,84]. The elderly should devise a strategy and expand their resources for mobility task performance. Following Nordbakke’s research, individual resources, contextual conditions, and strategies were interconnected in terms of how they expanded mobility capabilities with more development opportunities [56].

The WHO engaged older adults in focus groups to highlight advantages and barriers in eight urban areas worldwide, to create a guideline that encourages age-friendly cities [85]. An age-friendly city alters its structures and services to accommodate older people’s needs and abilities and promote active ageing. Thus, the neighbourhood should promote an active lifestyle for elderly adults. Basic mobility facilities, involving pedestrian walkways, elderly-friendly public transportation, proper road lighting, proximity to basic facilities and other associated factors, must also be considered.

Age-friendly cities should include silver industries. Usui explained how Japan encourages senior citizens to live independently. In Japan, the dependent age ratio approach determines older adults’ dependence level. The high number, which burdened the Japanese economy, motivated policymakers to address this social complexity. One alternative is the silver industry, which strives to create independent senior persons [83,86]. Such industries create older consumer-based products and services: care centres, activity centres, and private companion services.

Sustainable development goals (SDGs) aim to increase the quality of urban living and age-friendly cities that recommend a safe transportation system for older adults to remain independent and age in place. An intervention through collaborations between public and private transportation companies could establish a new partnership for older adults to enjoy safe mobility for subjective well-being [87]. Simple mobility activities involving shopping, visiting friends, recreation, and eating, also require secure companion services to improve elderly well-being. Although online shopping and food delivery can be directly impactful as an intervention for older people, tech-savvy assistants must be verified to protect elderly individuals from online scammers.

As one of the expanding services in the silver industry, companion services notably improve elderly mobility. A companion provides companionship, practical and emotional support, and improved social networks, confidence, and independence. Resultantly, older adults are encouraged to engage in their local community [88,89]. Transportation becomes easier with the presence of companions, who offer transportation services [90] and introduce the use of public transportation to older adults who are unfamiliar with them [89]. Companionship is also a solution for elderly people without a license or vehicle, or those who have given up driving, live alone, or want to depend less on children or friends. Most elderly people do not prefer to rely on family members or friends [91]. In this sense, companionship presents various beneficial implications and advantages in terms of psychology and lifestyle [88]. This phenomenon implicitly motivates and catalyses elderly mobility. A positive relationship was identified between older people’s engagement in physical activities and their overall well-being [15].

### 4.4. Limitation of the Study

This study encountered several limitations. Although the researchers identified some articles on how urbanisation affects low elderly mobility, the topic was not extensively covered. Only several stories emphasised the extent to which well-planned development could facilitate elderly mobility. Future works could extensively discuss these subject areas. Based on the search strategy involving four search engines, none of the studies examined the COVID-19 impact on elderly mobility. Most of the works were also conducted statistically, with the results focusing on mobility patterns and comparison studies, rather than elderly experiences, feelings, and thoughts. Hence, further research should investigate the impacts of urbanisation and the COVID-19 pandemic on elderly mobility.

## 5. Conclusions

In line with this scoping review, health issues primarily instigated low mobility and other associated barriers, such as fear of falling, the inability to use public transportation, and the need to be accompanied. Furthermore, the key factors influencing elderly mobility barriers could be divided into two categories: personal (health conditions, social relationships, and socio-economic background) and environment (built environment involving (i) facilities and structures and natural barriers, such as (ii) hills and weather factors). Older adults must undertake relevant measures, such as elderly companion services, to preserve their mobility and well-being. Meanwhile, policymakers should consider the WHO guidelines to develop age-friendly cities and silver industries. The rising number of elderly people must be seriously regarded to prevent negligence or discrimination. Overall, this study supports the SDGs for older adults in the following areas: (i) SDG3: Enable healthy ageing, well-being, and access to health and care services; (ii) SDG11: Build inclusive and accessible cities and communities.

## Figures and Tables

**Figure 1 ijerph-20-04243-f001:**
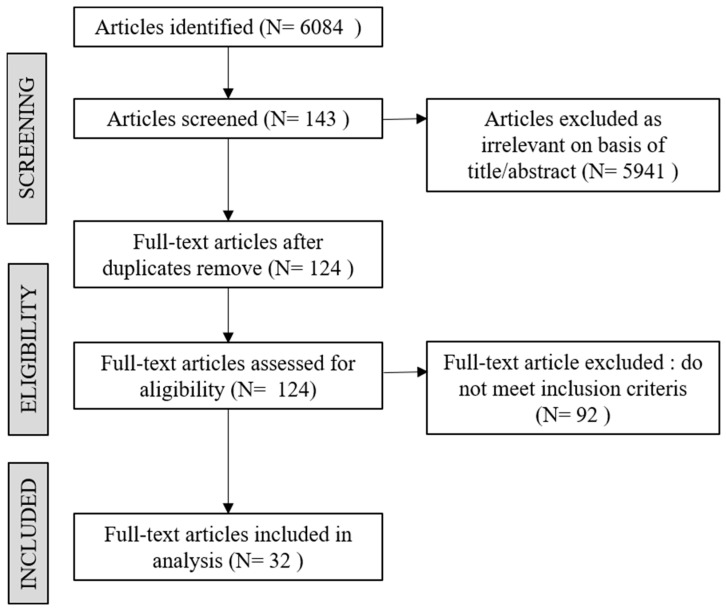
PRISMA process flow.

**Table 1 ijerph-20-04243-t001:** Mobility barrier theme, derived from included studies using TA.

Themes	Health	Build Environment	Socio-Economic Background	Social-Relation Changes	Weather
Sub-themes	Mobility impairment, illness, fear of falling	House location, hills, facilities, structure, public transport issues, urbanization	Education, income, license and car ownership	Living alone, losing spouse, losing family or close friends, moving to other location, retirement	Heavy rain/snow
Number of included studies	19	16	14	11	2
References	[11,25,39,40,41,42,43,44,45,46,47,48,49,50,51,52,53,54,55]	[11,16,42,43,44,46,50,51,52,56,57,58,59,60,61,62]	[11,22,23,39,41,42,43,46,47,56,60,62,63,64]	[11,25,39,43,46,49,51,53,54,55,65]	[16,52]

## Data Availability

Not applicable.

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
