# Peer review of "A Scoping Review of the Factor Associated with Older Adults’ Mobility Barriers"

_ijerph, 2023, doi:10.3390/ijerph20054243_

Round 1

Reviewer 1 Report (Previous Reviewer 2)

Thank you for resubmitting your paper.

It has been revised significantly from the previous paper. I think you have carefully responded to the comments from reviewer 1 in particular.

Thank you for your good work.

This manuscript is a resubmission of an earlier submission. The following is a list of the peer review reports and author responses from that submission.

Round 1

Reviewer 1 Report

This article identified factors associated with mobility barriers in older adults through a systematic literature review. I have here my main recommendation about this manuscript:

The authors provided a good introduction to mobility and its significance to well-being and independent life. However, the need for this review is not well justified – a more critical lens is needed to establish the need for the study. The authors stated that “The aim of this article is to focus on the barriers and challenges that older adults faced, resulting in mobility problems.” What is missing in the introduction is a problem statement highlighting why this review needs to be conducted. I would encourage the authors to identify previous review articles on this subject matter, identify the research gaps and highlight the significance of this review (i.e., how the proposed review fills the identified gap) and how this current review study is different from existing studies.

In the Abstract the authors mentioned they used four search engines but, in the Materials and Methods section the authors mentioned they used two electronic databases – please clarify this inconsistency. Also, why was the search limited to 2011-2022? Are articles published prior to 2011 not relevant? Your choice of keywords needs to be justified. The eligibility criteria for the title or abstract screening should be stated as well. Do not assume everyone knows what the PRISMA protocol is – provide background information on PRISMA and why it is being used in this review. I would encourage the authors to read about thematic analysis – this could be used in the Data Analysis section.

“We identified three types of articles: i) Quantitative research, and ii) Qualitative research. The quantitative method was used in 18 studies, while the qualitative method was used in 8 studies” What is the third type of article?

Line 161: “This structure is similar to Mollenkopf and Yeom et.al.” Provide full citation.

Line 164: “Yeom et al. addressed the intrapersonal …” Provide full citation.

Provide further justification for proposing a different categorisation of the barriers – in comparison to the studies you mention (i.e., Mollenkopf and Yeom et.al and Yeom et al.’s categorisation). You may support this proposition with existing theories. Also, why did you limit your categorisation to just Mollenkopf and Yeom et.al and Yeom et al? A good review should identify more of these studies and categorisation. The theoretical implications of the result should be highlighted in the discussion.

Several grammatical errors and typos in the manuscript.

Author Response

Dear Reviewer, 

Please refer to attachement.

Thank you.

Reviewer 2 Report

I appreciate you letting me read your very interesting review paper. However, I think this paper has some major problems.

I agree that the review is a scoping review, not a systematic review, and therefore the review will cover a wide range of articles. The reviewers understood the research question ("What are the factors associated with mobility barriers among older adults?"). However, the definition of "mobility barrier" should be clarified. In addition, I think that the content of the eligibility criterion b is very important. This point is unclear.

Reviewers should be specific about the relevance of the "mobility barrier" . By clarifying this point, the search keywords will become clearer. In the current paper, I think the procedure from the research question to the literature screening is unsuitable.

Is Figure 2 described by the author from this study? Has it already been described in previous studies?

Author Response

(The authors gave the same response as above.)
